# Seasonal modulation of deep slow-slip and earthquakes on the Main Himalayan Thrust

Dibyashakti Panda [1], Bhaskar Kundu[1], Vineet K. Gahalaut[2], Roland Bürgmann [3], Birendra Jha[4], Renuhaa Asaithambi[4], Rajeev Kumar Yadav[5], Naresh Krishna Vissa[1] & Amit Kumar Bansal[6]

The interaction between seasonally-induced non-tectonic and tectonic deformation along the Himalayan plate boundary remains debated. Here, we propose that tectonic deformation along this plate boundary can be significantly influenced by the deformation induced by the non-tectonic hydrological loading cycles. We explore seasonal mass oscillations by continental water storage in Southeast Asia and Himalayan arc region using continuous Global Positioning System measurements and satellite data from the Gravity Recovery and Climate Experiment. We suggest that the substantially higher transient displacements above the base of the seismogenic zone indicate a role of changes in aseismic slip rate on the deep megathrust that may be controlled by seasonal hydrological loading. We invoke modulation of aseismic slip on the megathrust down-dip of the seismogenic zone due to a fault resonance process induced by the seasonal stress changes. This process modulates mid-crustal ramp associated micro-seismicity and influences the timing of Central Himalayan earthquakes.

[1] Department of Earth and Atmospheric Sciences, NIT Rourkela, Rourkela 769008, India. [2] National Centre for Seismology, Ministry of Earth Sciences, New Delhi 110003, India. [3] Department of Earth and Planetary Science, University of California, Berkeley, Berkeley 97720-4767 CA, USA. [4] Department of Chemical Engineering and Materials Science, University of Southern California, Los Angeles 90007-1211 CA, USA. [5] Department of Earth Sciences, IIT-Kanpur, Kanpur 208016, India. [6] CSIR-National Geophysical Research Institute, Hyderabad 500007, India. Correspondence and requests for materials should be addressed to B.K. (email: rilbhaskar@gmail.com)

High relief of the Himalayan Mountains blocks the northward monsoonal winds from the Indian Ocean during June to September, resulting in intense precipitation on the southern slopes of the ranges[1]. This orography-controlled precipitation pattern, and associated seasonal water mass accumulation and mass redistribution by the Himalayan river system, dominates regional surface loading along the arc and in the adjoining Indo-Ganga plains. Also, during the winter period snow accumulates in glaciers on high peaks of the Himalaya. These seasonal hydrological mass movements influence secular deformation of the lithosphere and possibly impact tectonic processes in the Central Himalaya and adjacent regions[2] (Fig. 1a, b). Taking advantage of continuous Global Positioning System (cGPS) measurements and satellite data from the Gravity Recovery and Climate Experiment (GRACE), several investigators[3,4] have analyzed seasonal variations of continental water storage and its temporal evolution in Southeast Asia. However, the exact interaction between seasonally-induced non-tectonic and tectonic deformation along the Himalayan plate boundary remains elusive (Fig. 1b). Moreover, the sensitivity of potential stress triggering relationships to the fault geometry and the possible role of rate-and-state-dependent friction effects challenge a clear demonstration of the effect of periodic mass oscillations on the nucleation process of Himalayan earthquakes. In this letter we focus on this issue, which has implications for Himalayan tectonics, seismic hazard, and seasonal deformation of Southeast Asia.

Along the orogenic boundary of the Himalayan arc, the convergence of ~20 mm year$^{-1}$ is mostly accommodated by slip on the plate interface, referred to as the Main Himalayan Thrust (MHT), which has hosted several large historic megathrust earthquakes, including the recent 2015 Mw 7.8 Gorkha earthquake[5,6]. GPS measurements in the Himalayan region clearly show evidence of convergence and elastic strain accumulation[7]. However, in addition to the secular motion, there are strong seasonal variations in the motion, most prominent in the north and vertical components[3,4,8]. In addition to rare, large-magnitude plate-boundary events, interseismic strain accumulation in the Himalaya is also associated with micro-seismicity occurring close to the mid-crustal ramp of the MHT, which shows evidence of seasonal periodicity in response to annual stress variation due to hydrological loads[2,9–11]. Visual inspection of time series of GRACE-derived equivalent water height (EWH), regional rainfall, historic earthquakes, current seismicity (of M > 5) and corresponding seismic moment release, and micro-seismicity (see Methods), suggests that the earthquake occurrences have some correlation with the timing of the hydrological loading cycle in the Nepal Himalaya (Fig. 2). Our cross-correlation analysis among the various physical parameters (Supplementary Figs. 1, 2, 3, 4 and 5) confirms that the phase relationship between

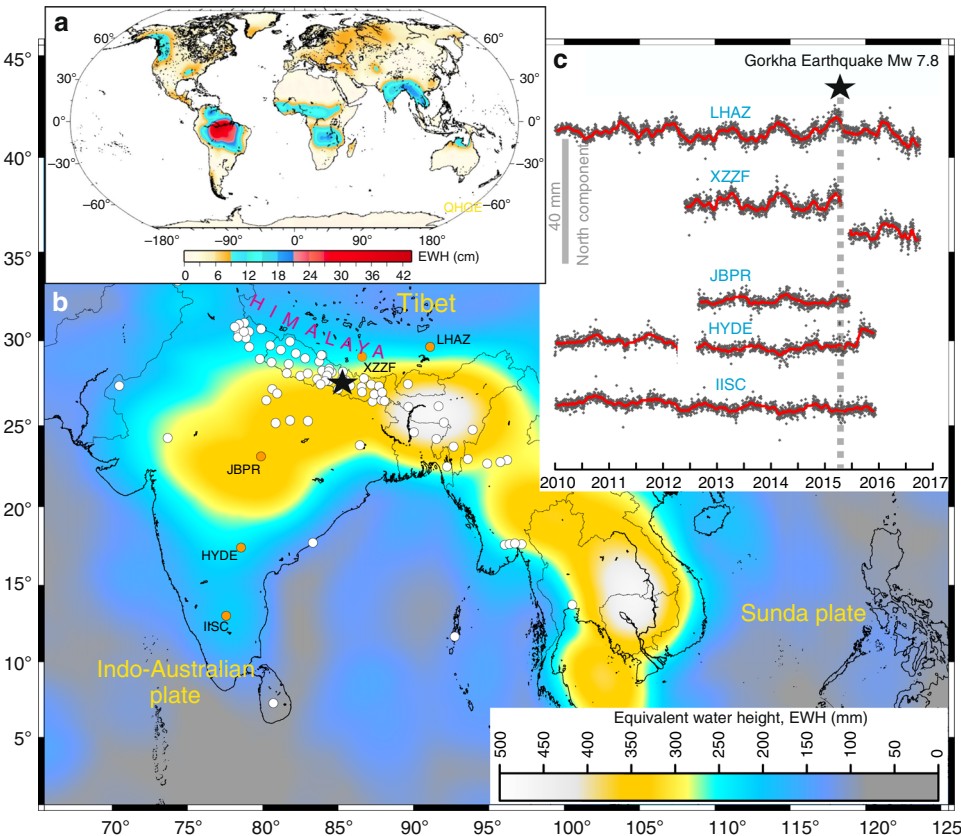

**Fig. 1** Seasonal deformation of Southeast Asia and its influence over tectonic deformation along the Himalaya. **a** Global seasonal mass variation of equivalent water height (EWH, in cm) measured by GRACE[3]. The Amazon basin and Southeast Asia experience large seasonal mass variations. **b** Average peak-to-peak annual surface load variations over Southeast Asia, in terms of equivalent water height (EWH, in mm) measured by GRACE for the period 2002–2012. Circles show the location of the cGPS sites from different networks in the Southeast Asia region. Epicenter of April 25, 2015 Mw 7.8 Gorkha Earthquake is shown by the black star. **c** Small inset represents detrended cGPS time series (north component) of five representative sites from China (stations LHAZ and XZZF) and India (stations JBPR, HYDE, and IISC). It is evident that seasonal deformation occurred both before and after the 2015 Gorkha earthquake (marked by dashed line). The map in (**a**) is reused from Fu et al.[3], with permission from John Wiley and Sons. All rights reserved. The map in (**b**) was created by authors using Generic Mapping Tools, version 5.2.1; URL: http://gmt.soest.hawaii.edu/

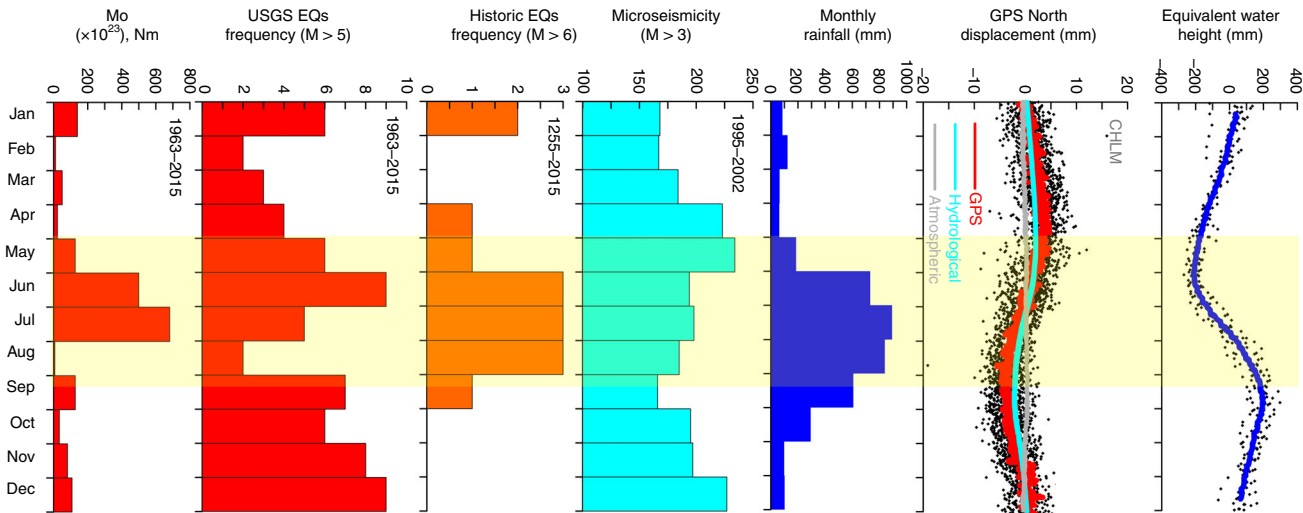

**Fig. 2** Correlation between equivalent water height (EWH), rainfall, GPS displacement, and different seismicity catalogues. Stacked panels of annual changes of equivalent water height (derived from GRACE averaged over the region latitude: 27.5°–20°N, Longitude: 75°–88°E region), representative GPS displacement in north component (station CHLM), monthly rainfall in Nepal and surrounding region, micro-seismicity from Nepal (1995–2002 of M > 3), historic earthquake frequency (1255–2015 of M > 6), current earthquake frequency from USGS (1963–2015 of M > 5), and USGS catalog seismic moment (Mo, Nm). We suggest that the timing of Nepal Himalayan earthquakes of all magnitudes (historic earthquakes of M > 6, recent earthquakes of M > 5, micro-seismicity of M > 3) is influenced by seasonal deformation

hydrological loading and precipitation is complex, as much of the water mass is temporarily stored in surface reservoirs and groundwater thus delaying (~1.5 months) its transport out of the system. Further, GRACE-derived EWH and regional precipitation exhibit strong lag-correlation with the occurrence of historic earthquakes, micro-seismicity, and seismic moments of earthquakes in the USGS catalog. However, the hydrological load-induced modulation of interseismic strain accumulation, the physics of the associated earthquake occurrences, and mechanical role of the Himalayan ramp/flat segments along the MHT remain poorly understood[12].

## Results and discussion

**Seasonal transients in GPS time series.** Bettinelli et al.[2] first demonstrated how the seasonal variation in water loading induces deformation in Nepal Himalaya. A point load located on the Earth's surface causes the nearby region to subside and to move horizontally towards the load. The pattern is reversed during an unloading episode (Fig. 3). Although the geometry of the hydrological load over Southeast Asia is complex (Fig. 1b), the cGPS sites located in the Nepal and Kumaun-Garhwal Himalaya show similar seasonal deformation behavior. On average, the north component of the annual cGPS position time series, which is aligned with the predominantly northward direction of India-Eurasia convergence, reflects seasonal displacement of the sites associated with the hydrological loading. Here, we focus on these seasonal cycles by removing the secular motion from the time series. It appears that during the pre-monsoon period (October–April) sites move northward while during the monsoon period (May–September), sites along the Himalaya experience southward movement (Fig. 1c)[3,4,8]. Stations to the south of the Plains show the opposite pattern of seasonal motion (see station HYDE and IISC in Fig. 1c). There are mismatches in the amplitude and phase of seasonal transients with the expected hydrological load signal at a few sites in southwest Nepal (e.g., Dhangadi—DNGD, Ghorahi—GRHI, Nepalganj—NPGJ) and Kumaun-Garhwal Himalaya (e.g., Kunair—KUNR, Raithal—RATH) due to local effects of mountain valley, hydrology, groundwater irrigation, and reservoir impoundments[3,13].

We fit a hyperbolic tangent function (HTF) (see Methods) to the cGPS time series, to quantify the seasonal transients in the Nepal and Kumaun-Garhwal Himalaya, and project them with increasing distance from the Main Frontal Thrust (MFT) in the NNE direction (Fig. 4a). The sites in the Nepal Himalaya have higher seasonal transients than the sites located in the Kumaun-Garhwal Himalaya implying a stronger effect of the hydrological load over the Nepal region. The effect of hydrological loading is also more prominent in central and eastern Nepal compared to western Nepal. Since the intensity of monsoonal rainfall and the related GRACE-derived load signal increases across India from west to east, the seasonal deformation also increases to the east. Seasonal transients are especially high with respect to those predicted from the load model derived from the GRACE time series (see Methods) at sites located over the mid-crustal ramp and adjacent flat of the MHT (Fig. 4a, b and Supplementary Fig. 6). These sites lie about ~70–110 km north of the MFT[12] above the base of the seismogenic zone (Supplementary Fig. 6).

The horizontal displacements induced by hydrological loading are generally less than about half of the vertical displacements, with variations introduced by the elastic Earth structure[8,14]. However, the observed ratio between horizontal and vertical transients is often significantly higher than 0.5, and twice exceeds 1.0, for sites that are located over the ramp and adjacent flat. The horizontal transient displacements decay with distance away from the base of the seismogenic zone (Fig. 4a). Abnormally high seasonal transients at these sites are in excess of model predictions from hydrological and atmospheric loading (Fig. 4b, Supplementary Fig. 6, see Methods). The amplitude and phase of the seasonal transients of horizontal geodetic positions (i.e., in north component) cannot be predicted as the response of a spherical and layered elastic Earth to annual variations of continental water storage, as suggested in the previous work[8]. However, it has been suggested that the fit to the geodetic data can be improved by adjusting the layered Earth model[8]. We argue that the substantially higher transient displacements above the base of the seismogenic zone suggest a role of changes in aseismic slip rate on the deep megathrust that may be controlled by the seasonal hydrological deformation. This motivates us to further

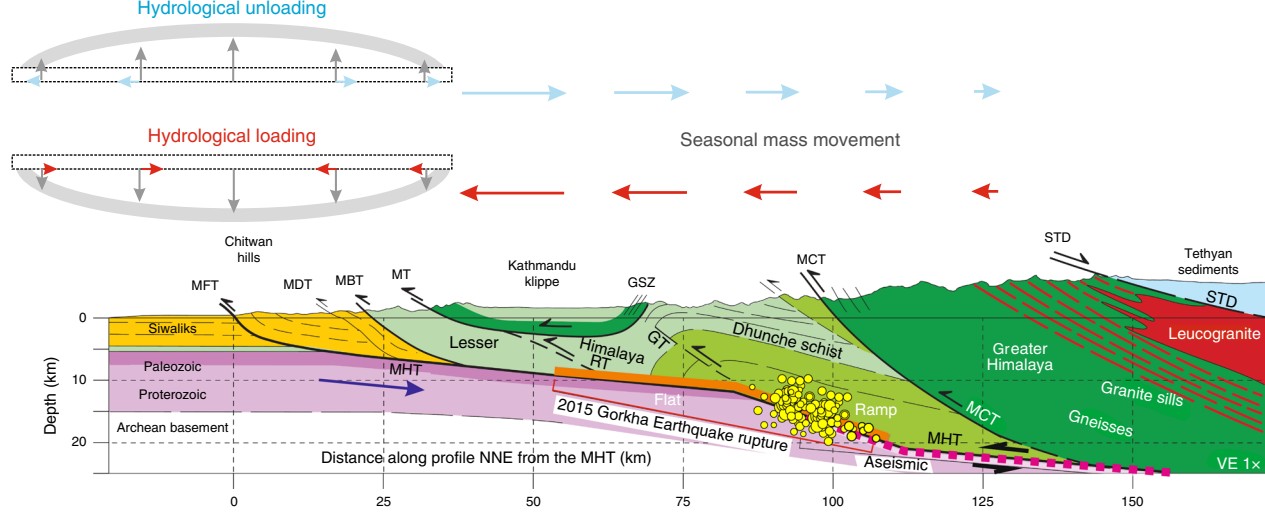

**Fig. 3** Conceptual model describing the effect of the hydrological mass oscillation (loading/unloading) along the North–South profile[6] of Nepal-Himalaya region. Hydrological loading in the Indo-Ganga plains causes subsidence and southward movement in the Himalayan region. GSZ Galchi shear zone, GT Gorkha blind Thrust, MHT Main Himalayan Thrust, STD South Tibetan Detachment, MCT Main Central Thrust, MT Munsiari/Mahabharat Thrust, RT Ramgarth Thrust, MBT Main Boundary Thrust, MDT Main Dum Thrust, MFT Main Frontal Thrust

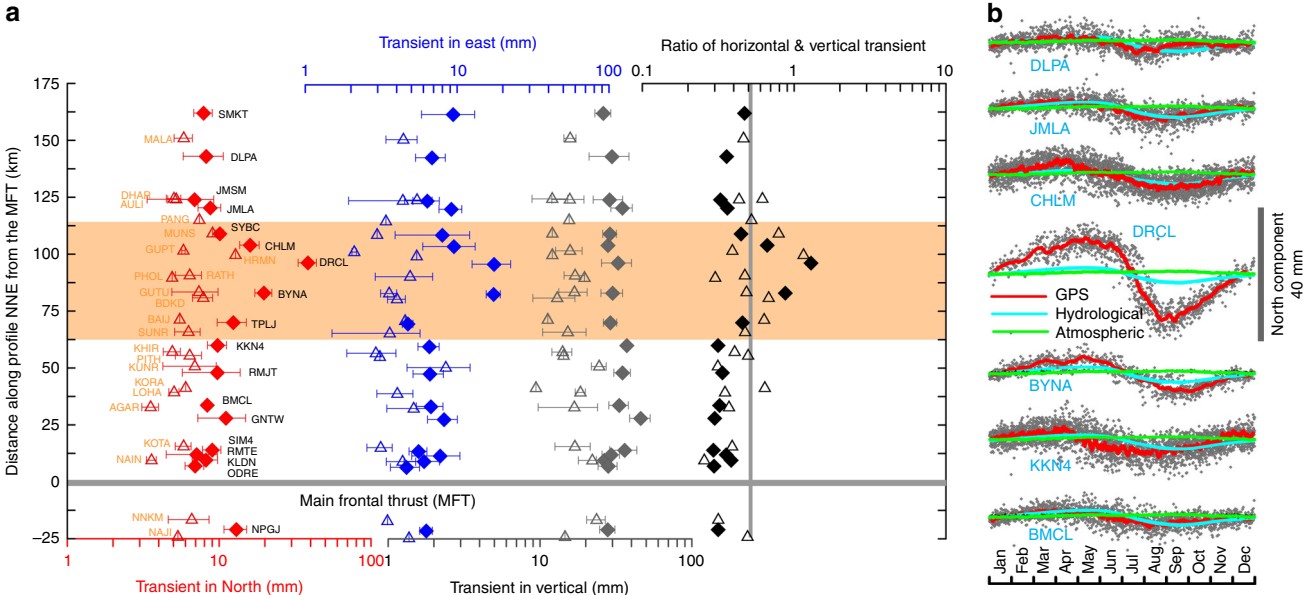

**Fig. 4** Evidence of seasonal transient deformation in cGPS networks from the Nepal and Kumaun-Garhwal Himalaya. **a** cGPS-derived mean seasonal transients (in North, East, vertical, and ratio of horizontal and vertical) are projected along a NNE profile with distance measured from MFT, along with their 1σ error bars. Filled and open symbols represent data from the cGPS networks in Nepal and Kumaun-Garhwal Himalaya, respectively. Transient deformation is abnormally higher in the region of the mid-crustal ramp and adjacent updip segment (marked by horizontal orange color strip). **b** North component of average seasonal deformation measured by cGPS and simulated deformation due to hydrological and atmospheric loading in the Nepal Himalaya. Note the sites BYNA, DRCL, CHLM show transients in excess of hydrological and atmospheric model prediction

probe the nature of the mechanical interaction of the Himalayan ramp/flat segments and associated seismicity in response to seasonal deformation. We first show that the influence of the hydrological load, in terms of change in stress, is stronger on the ramp (located close to the base of the seismogenic zone) compared to on the updip flat of the MHT. We argue that these perturbations in stress cause large displacement transients due to the fault resonance effect.

**Stress change on mid-crustal ramp and flat**. We use GRACE satellite data to determine the seasonal peak-to-peak surface load

variations in EWH in Southeast Asia and adjacent regions (Fig. 1b, see Methods). We use a coupled flow and geodynamics simulation framework[15] to calculate the effect of time-dependent hydrological surface load variations on the MHT. Quasi-static equilibrium between gravity, tectonic stresses, and pore pressure is applied over the 3D model domain under reasonable boundary conditions. We specify zero slip on the MHT, and calculate the changes in pore pressure and stress state as a result of the surface load-induced pressure diffusion and poroelastic stress changes over the period of 2002–2012. We solve the coupled fluid flow and mechanical deformation problem and do not assume

undrained deformation. Finally, hydrological load-induced shear and effective normal stress components are projected on the receiver fault segments, with a focus on the mid-crustal ramp and flat, of the MHT (see Methods, Supplementary Figs. 7 and 8). Figure 5a shows hydrological load-induced peak-to-peak shear, normal and Coulomb failure stress changes (considering an effective friction coefficient of 0.40) on the mid-crustal ramp and adjacent flat. It reveals several important features of seasonal stress perturbation on the Himalayan detachment: first, slip encouraging shear and normal stresses are in phase and also correlated with the northward displacement variations. Second,

shear-stress perturbations are about three times larger on the mid-crustal ramp than on the updip flat segment; however, normal-stress perturbations are slightly higher on the flat than on the ramp. Computed Coulomb stress perturbations are also higher on the mid-crustal ramp than on the flat, suggesting that stress variations are sensitive to the fault dip. Third, such higher shear stress focusing on the mid-crustal ramp can possibly explain the abnormally higher seasonal transient in cGPS displacement at sites near the base of the seismogenic zone, thereby supporting the possible role of frictional dynamics and control by the seasonal stress variations.

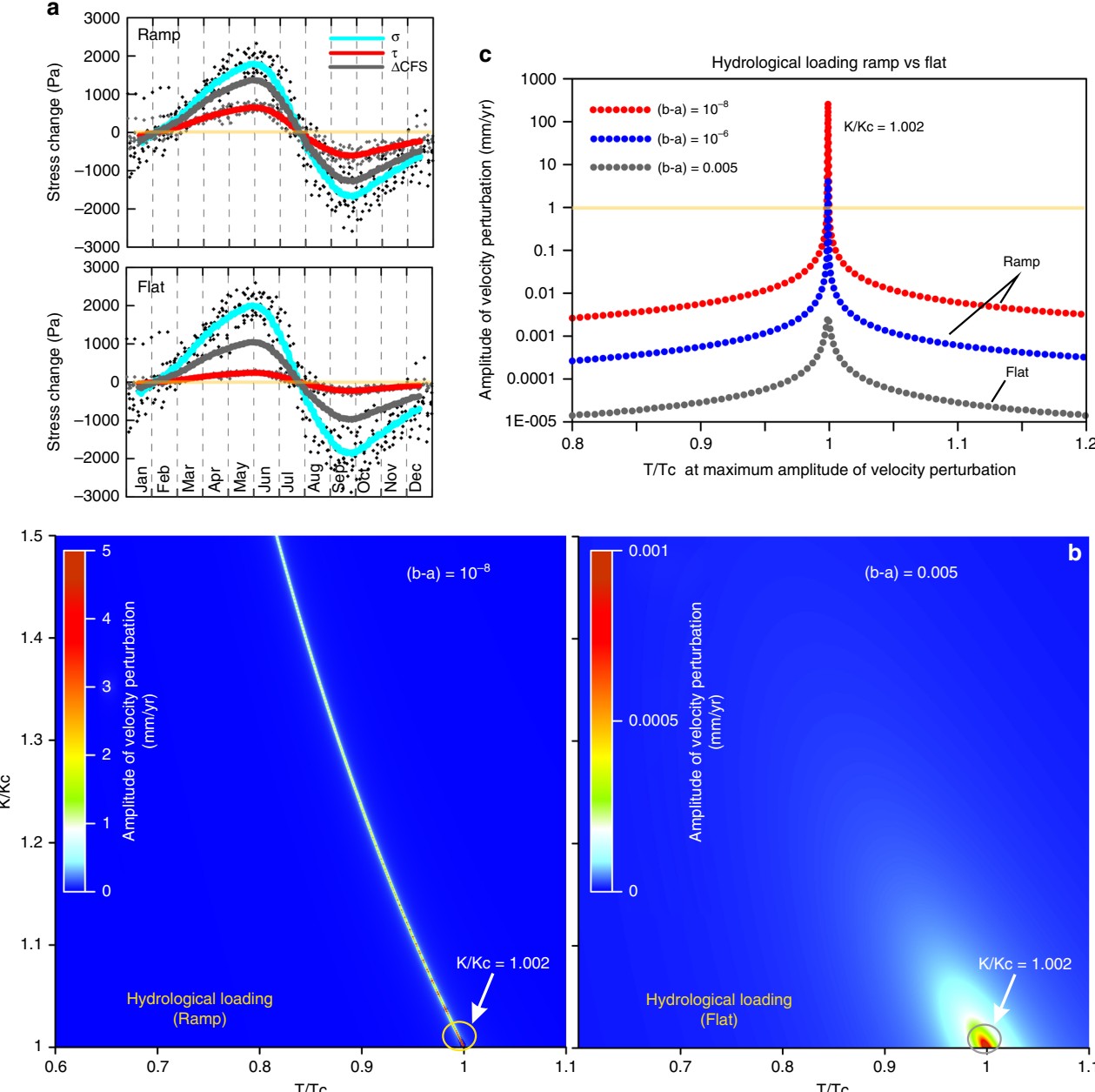

**Fig. 5** Hydrological load-induced Himalayan fault resonance. **a** Stacked stress profile on the MHT (ramp and flat). Effect of surface hydrological load variations (see Methods) on the mid-crustal ramp (at 15–25 km depth with dip as 20°) and adjacent Himalayan flat (at 15 km depth with dip 7°) along the MHT. Hydrological surface load-induced shear (red; positive favors thrust slip) and normal and Coulomb stress changes (light blue, negative indicates fault compression) are also shown. **b** Relationships of rate- and state-dependent frictional parameters to resonant mid-crustal Ramp and Flat (MHT segments) during hydrological loading episodes. Slip velocity perturbation as a function of period $T$ and fault stiffness $k$, for specific frictional parameters. **c** Amplitude velocity perturbation approaches its maximum for $k/k_c = 1.002$. Note the amplitude of velocity perturbation is several orders of magnitude higher on the mid-crustal ramp than on the flat along MHT during the time of hydrological loading episodes

**Fault resonance effect on MHT and its implication**. It has been proposed that a creeping fault can be destabilized and can enter into a stick-slip domain by periodic stress oscillations[16]. We adopt the approach of Perfettini et al.[16] to better understand the role of frictional dynamics on the Himalayan ramp and flat, controlled by the seasonal stress variations (Fig. 5a, see Methods). Figure 5b shows the calculated slip velocity ($V$) perturbation with respect to frictional conditions as a function of stiffness ($k$), period ($T$) and stress change (Fig. 5a, see Fault resonance model in Methods). In these computations we used a critical slip distance $D_c$ of $10^{-6}$ m, critical period $T_c = 1$ year, steady-state friction $\mu_{ss} = 0.3$, and India-Eurasia relative plate motion velocity $V_{rpm} = 20$ mm year$^{-1}$, which are reasonably compatible with laboratory estimates of frictional parameters[16–18] and seismo-tectonic conditions of the central Himalaya[7]. Although low values of ($b - a$) of the order of $10^{-4}$ for velocity-weakening behavior have been found experimentally[19], there appears to be no lower threshold on this estimate, as long it is considered positive and non-zero[20]. We assumed very small but positive values ($\sim 10^{-8}$ and $10^{-6}$)[17] for ($b - a$) for the base of the Himalayan seismogenic zone (i.e., ramp), while at the intermediate depths of the seismogenic zone (i.e., on the flat), we considered ($b - a$) as 0.005. Similar estimates for ($b - a$) have been proposed elsewhere[17–20] and are also compatible with laboratory estimates of frictional parameters[16,18–20]. In the ramp region, the ratio $a/b$ corresponds to $>0.9$[19]. It has been suggested[17] that the slip pulse is significantly amplified for the narrow range of periods near $T_c$ when $k = k_c$ (i.e., for shorter periods when $k > k_c$, where $k_c$ represents critical stiffness). The amplitude of the velocity perturbation approaches its maximum for $k/k_c = 1.002$ (Fig. 5c). It appears that slip velocity perturbation is several folds higher on the mid-crustal ramp segment than on the flat during hydrological loading episodes suggesting that the Himalayan ramp is more sensitive to seasonal modulation than the updip flat (Fig. 5b, c). Although such stress perturbations ($\sim 2$ kPa) are a negligible fraction of the stress drop during large inter-plate earthquakes on the Himalayan plate boundary ($\sim 0.02\%$), they may be sufficient to resonate the MHT to produce resonant slip accelerations on the MHT downdip of the seismogenic zone[17]. Lowering of effective normal stress (or increasing pore fluid pressure) can also lead to an increase in the amplitude of the velocity perturbation (i.e., fault resonance process, Supplementary Fig. 9). Therefore, the effect of the fault resonance process can also be influenced by the presence of fluids[21,22], in response to variations in pore fluid pressure.

The rate-and-state frictional model also provide scaling arguments for faults to slip by the resonance process caused by periodic stress oscillations[19]. It has been suggested that the half-length of the minimum available patch size, $h^{\star}$, required to nucleate such slip events is given by[23,24]: $h^{*} = \frac{G'D_c}{(b-a)\sigma_e}$, where, $h^{\star}$ is similar to parameter $h_c$ proposed by Perfettini et al.[16]. Here, $G'$ is the effective shear modulus $\left(\frac{G}{1-\vartheta}\right)$, $G$ is the shear modulus, and $\vartheta$ is Poisson's ratio. Considering the values of $G = 30$ GPa, $\vartheta = 0.25$, $b - a = 10^{-6}$, critical slip distance ($D_c$) = $10^{-6}$ m and effective normal stress ($\sigma_e$) = 300 MPa for the Himalayan ramp segment, the minimum patch size ($h^{\star}$) required to nucleate a slip event is estimated as $\sim 0.133$ km. However, upon decreasing the effective normal stress to 30 MPa (via a corresponding increase in pore fluid pressure), $h^{\star}$ is increased to $\sim 1.33$ km. For slow slip governed solely by rate-and-state friction to remain stable, the fault length must be smaller than $2L_{\infty}$[25], where $L_{\infty} = \frac{G'D_c b}{\pi \sigma_e (b-a)^2}$. Now, with the above mentioned parameters, $L_{\infty}$ is found to be $\sim 424$ km, assuming effective normal stress ($\sigma_e$) as 300 MPa. $L_{\infty}$ will increase further with decreasing effective normal stress (or increasing pore fluid pressure). Finally, it has been suggested that

the condition required for a portion of the fault to slip through fault resonance process should follow $2h^{\star} < W < 2L_{\infty}$[19], where $W$ is the length of the slow-slip patch defined by: $W = \frac{u_{SSE}G'}{\beta \sigma_e (b-a) \ln\left(\frac{V_{SSE}}{V_{creep}}\right)}$, where $\beta$ is an empirical constant of the order of 1, $u_{SSE}$ is slip amount, and $V_{SSE}$ is the slip speed during the slow slip event. However, in order to investigate the fault resonance process in the Himalaya, it is difficult to constrain the exact values of $u_{SSE}$ and $V_{SSE}$. Therefore, considering that limitation, if we assume that the whole length of Himalayan mid-crustal ramp is resonating, then in that case $W$ would be $\sim 25$ km. This indicates that the above scaling condition (i.e., $2h^{\star} < W < 2L_{\infty}$) for nucleation of slow-slip events and associated slip perturbations is generally consistent for the Himalayan ramp region.

Therefore, the slip resonance may explain why transient-slip and abnormally higher transient in cGPS displacement should be favored near the transition from velocity-weakening to velocity-strengthening behavior at the base of the Himalayan seismogenic zone (i.e., on the mid-crustal ramp). Thus, fault resonance hypothesis, implying change in slip rate on the ramp, appears as a suitable working hypothesis for seasonal modulation in the Nepal and adjacent Kumaun-Garhwal Himalaya. Further, May–June peak in Coulomb stress perturbations on the MHT is consistent with the slow-slip acceleration helping drive distributed seismicity in the hanging wall on the thrust. We conclude that the tectonic deformation along this plate boundary is influenced by non-tectonic seasonal stress perturbations, which ultimately influences the timing of central Himalayan earthquakes.

## Methods

**GPS data and processing**. In 2012–2013, a network of 23 cGPS sites were installed in the Kumaun-Garhwal Himalaya to monitor crustal deformation and strain accumulation along this part of the Himalayan arc. Continuous GPS data from these stations, along with the Nepal geodetic network (Fig. 1), have been analyzed together with several IGS sites surrounding the Indian plate. The Nepal network consists of three cGPS stations, which were installed in 1997 under collaboration between the Laboratoire de Détectionet Géophysique (CEA/LDG, France) and the Department of Mines and Geology (DMG, Nepal), and 25 stations which were deployed since 2003 by the Tectonics Observatory (http://www.tectonics.caltech.edu). We used GAMIT, version 10.60[26–28], to estimate the GPS time series of site coordinates and their mean velocities. Site position estimates and their rates were estimated in ITRF2008[29] by stabilizing sites in stable continental regions and IGS reference sites using GAMIT/GLOBK. Processed cGPS time series data from the Tibetan plateau can be accessed at ftp://ftp.cgps.ac.cn/.

**Seasonal transient estimates in cGPS time series**. We use a deformation modeling tool[30] that fits a HTF to the cGPS time series to evaluate and quantify seasonal transients (Fig. 4a). Seasonal displacements can be estimated by fitting the cGPS time series with a function in the form of:

$$\vec{\mathbf{x}}(\mathbf{t}) = \vec{\mathbf{x}}_0 + \vec{\mathbf{V}}t + \sum_{i=1}^{n} \frac{\vec{\mathbf{U}}_i}{2}\left[\tanh\left(\frac{t - T_{0i}}{\tau_i}\right) - 1\right] \quad (1)$$

where $\vec{\mathbf{x}}(\mathbf{t})$ are cGPS site coordinates at time $t$, $\vec{\mathbf{x}}_0$ are coordinates at a reference time, $\vec{\mathbf{V}}$ is "steady-state" velocity, $\vec{\mathbf{U}}_i$ is the anomalous seasonal displacement during the $i$th of $n$ seasonal events, $T_{0i}$ is the median time of the $i$th seasonal event, and $\tau$ scales the period over which the event occurred. If $T_0$ and $\tau$ are specified, the other parameters can be estimated from linear least-squares inversion and an algorithm for grid search over $T_0$ and gradient search over to estimate anomalous seasonal deformation events is available at http://aconcagua.geol.usu.edu/~arlowry/code_release.html.

**EWH computation from GRACE hydrological model**. Geodetic horizontal (North and East components) and vertical (Up component) seasonal displacements captured by cGPS can be predicted as the response of a spherical and layered elastic Earth to annual variations of continental water storage[8]. We used GRACE to determine the seasonal time evolution of surface loading (i.e., peak-to-peak annual surface load variations) and EWH in Southeast Asia and adjacent regions (presented in Fig. 1). We compute surface displacements induced by loading using a layered non-rotating spherical Earth model[31] based on the Preliminary Reference Earth Model (PREM) and a local seismic velocity model[32,33]. In this computation, we used 10-day solutions of GRACE data provided by the Centre National d'Etudes

Spatiales/Groupe de Recherche de Géodésie Spatiale (CNES/GRGS), available at http://grgs.obs-mip.fr/. The solutions are presented in terms of non-dimensional spherical harmonic coefficients of the geopotential called Stokes coefficients, which represent the gravitational effects mainly related continental hydrology[8]. The hydrological modeling software package used in this computation can be accessed at http://web.gps.caltech.edu/~software.html/software.html.

**Atmospheric and hydrological loading estimates**. We used the data and programs provided by the Global Geophysical Fluid Center (GGFC)[34,35] (http://geophy.uni.lu/ggfcatmosphere/ncep-loading.html) for the estimation of surface displacements at GPS sites due to atmospheric pressure loading. GGFC provides 6-hourly, global surface displacements at 2.5° × 2.5° resolution, derived from National Center for Environmental Protection's (NCEP) reanalysis of surface pressure. For the estimation of surface displacement due to hydrological load, we used the data and program provided by GFZ (ftp://ig2-dmz.gfz-potsdam.de/LOADING/HYDL). The hydrological load is taken from the hydrological Land Surface Discharge Model (LSDM), which includes daily estimates of soil moisture, snow and surface water mass in rivers and lakes on a regular grid 0.5° × 0.5°[36,37]. Hydrologically induced elastic surface deformation is calculated by convolving Farrell's Green's function with modeled hydrological mass distributions from the LSDM. The elastic deformation has been computed in the centre of Earth's frame (cF) on the basis of load Love numbers given for the elastic Earth model "ak135"[38].

**Stress model on the Himalayan Ramp and Flat**. We construct a coupled flow and geo-mechanical model of the MHT region to infer the stress conditions influenced by the hydrological load over the Indo-Ganga Plain. In our model, we use the EWH data from GRACE mission to apply tractions on the ground surface and simulate the hydrological load-induced loading/unloading over a period of 10 years (2002–2012). We investigate the evolution of stress on seismogenic fault segments under different scenarios of surface loading-unloading and tectonic loading around the fault sections.

The computational domain of our finite element model implementation is a 3D box with the dimensions of 400 km × 40 km × 30 km in x, y, and z directions, respectively, where the y-axis is aligned with the north direction (Supplementary Fig. 7). The fault has three segments with depth-flat, ramp and aseismic flat, extending from a depth of 5–30 km. The characteristics of the fault segments are shown in Table 1.

We construct a tetrahedral mesh inside the domain with approximately 20,400 tetrahedral cells. Cell sizes vary in the domain with smaller cells close to the fault and coarser cells near the model boundaries. A 3D view of the mesh and the fault is shown in Supplementary Fig. 7. We assume a homogeneous isotropic poroelastic medium with its properties listed in Table 2. The model was initialized under thrust faulting conditions. The east and south boundaries are subject to compression that increases linearly with the lithostatic gradient. The west and north boundaries are roller boundary conditions. The top boundary is subject to the time-dependent hydrological load obtained from the EWH data.

We used our in-house coupled flow and geo-mechanical simulator that is based on PyLith (www.geodynamics.org/cig/software/pylith/) to model the effects of MHT geometry and seasonal hydrological loading on tractions on the MHT. Dynamics of the model is driven by the time-dependent surface load calculated from the EWH data. We specify zero slip on the fault and calculate the change in tractions on the fault as a result of the surface load. Quasi-static equilibrium of forces between gravity, total stress, and pore pressure under the prescribed boundary tractions provide the state of stress in the entire domain. The state

of stress around the fault is projected on the fault to obtain the effective normal and up-dip shear tractions on the fault segments as shown in Supplementary Fig. 8.

**Fault resonance model**. Perfettini et al.[16] proposed that a creeping fault can destabilize and enter into a stick-slip domain by periodic stress oscillation, which requires specific parameters to cause fault resonance. It is proposed that slip response to a given stress can be amplified within a narrow band of resonant periods, in that situation excitation by stress at period $T$ with shear stress ($\tilde{\tau}$) and normal stress ($\tilde{\sigma}_n$) amplitude perturbs the slip velocity, which can be expressed as[39]:

$$\tilde{V} = qV_{rpm}\frac{q\left[\tilde{\sigma}_n\left(\mu_{ss} - \alpha\right) - \tilde{\tau}\right] - i(\tilde{\sigma}_n - \tilde{\tau})}{(kD_c - a\sigma_n q^2) + iqD_c(k - k_c)} \quad (2)$$

$$q = \frac{2\pi D_c}{TV_{rpm}}, \quad (3)$$

$$K_c = \frac{\sigma_n(b - a)}{D_c} \quad (4)$$

where, $q$ is the dimensionless pulsation, $V_{rpm}$ is relative plate motion velocity, $\mu_{ss}$ is steady-state friction, $\alpha$ is the frictional response to change in normal stress, $k$ is the fault elastic stiffness ($k = \frac{\tau}{u} > k_c$), and $u$ is the slip deficit. Slip is resonant at critical stiffness (i.e., at $k = k_c$) and this occurs at critical period:

$$T_c = \frac{2\pi D_c}{V_{rpm}}\sqrt{\frac{a}{(b - a)}} \quad (5)$$

Figure 5b depicts the calculated slip velocity ($\tilde{V}$) perturbation with respect to frictional conditions as a function of stiffness $k$ and period $T$. In this computation, we used a critical slip distance $D_c$ of $10^{-6}$ m, $T_c = 1$ year, $\mu_{ss} = 0.3$ and an India-Eurasia relative plate motion velocity $V_{rpm}$ of 20 mm year$^{-1}$, which are reasonably compatible with the Nepal Himalaya. In this model, the slip pulse is amplified significantly for a narrow range of periods near $T_c$ when $k = k_c$ (i.e., for shorter periods when $k > k_c$). The amplitude of the velocity perturbation approaches its maximum for $k/k_c = 1.002$ (Fig. 5b).

**Rainfall, earthquakes, declustering, and Mc determination**. Precipitation data used in this paper can be found at Global Precipitation Climatological Centre (GPCC, http://www.esrl.noaa.gov/psd/). Micro-seismicity data from the National Seismic Centre of Nepal has been provided by J.-P. Avouac. The epicentre and the earthquake focal mechanism of 2015 Gorkha earthquake have been taken from the USGS National Earthquake Information Centre (NEIC). We have generated an aftershock-depleted micro-seismicity catalogue using the approach from Reasenberg[40], with $P = 0.95$, $1 \leq \tau \leq 10$ days, assuming a horizontal location error of 5 km and vertical error in hypocenter location of 10 km. We analyzed Gutenberg–Richter relation [log$N$ ($M \geq$ Mc) $= a - b \times$ Mc] of the aftershock-depleted catalog of the region during the period from 1994 to 2002, using maximum likelihood approach[41]. The b-value of the region during that period is 0.90 ± 0.03 for Mc = 2.1. Mc also shows slight annual variation, ~1.9–2.0 during winter and ~2.1–2.2 during summer. We use a conservative minimum magnitude of $M =$ 3. In the recent seismicity catalogue from the Nepal Himalaya (from 1963 to April 24, 2015, i.e., just a day before the April 25, Gorkha event), the b-value is 0.92 ± 0.05 with a Mc of 5.0.

## Data availability
All relevant data are available from the authors upon reasonable request.

**Table 1 Dip and depth interval of different segments of MHT in the geo-mechanical model**

| MHT segment | Dip (degree) | Depth interval (km) |
|---|---|---|
| Flat | 7 | 5–15 |
| Ramp | 20 | 15–25 |
| Aseismic zone | 5 | 25–30 |

**Table 2 Poromechanical properties of the study**

| | | | |
|---|---|---|---|
| Solid density | 2600 kg/m$^3$ | Porosity | 0.1 |
| Vp | 5900 m/s | Permeability | 500 mDarcy |
| Vs | 3400 m/s | Fluid density | 1037 kg/m$^3$ |
| Drained bulk modulus | 48.49 GPa | Fluid viscosity | 0.31 cP |
| Biot coefficient | 1 | Horizontal to vertical compression ratio | 1.1 |

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

## Acknowledgements

We appreciate the efforts of the scientists and technical staff of CSIR-National Geophysical Research Institutes, Hyderabad in maintaining the GPS network in India and Kumaun-Garhwal Himalaya. We thank J.-P. Avouac for sharing micro-seismicity data from the National Seismic Centre of Nepal. We thank J.-P. Avouac, Michael Steckler, Tony Lowry, and Kristel Chanard for stimulating discussion and suggestions. R.B. acknowledges support from the NASA ESI program. D.P., R.A., and R.K.Y. are supported by NITR Research Fellowship, USC Research Assistantship, and CSIR Fellowships, respectively. GPS network in the Kumaun-Garhwal region is financially supported by the Ministry of Earth Sciences, Government of India, through grant number MOES/PO/ (Seismo)/1(116)/2010.

## Author contributions

B.K., V.K.G., and R.B. provided research idea. D.P. performed cGPS data analysis and associated estimates of seasonal transients. B.K., V.K.G., and R.B. wrote the manuscript. B.K., R.A., and B.J. performed hydrological load-induced stress modeling. D.P., N.K.V., and B.K. performed Fault resonance model. R.K.Y. processed cGPS data from Kumaun-Garhwal Himalaya. A.K.B. involved in GPS data acquisition and maintaining the GPS network in India and Kumaun-Garhwal Himalaya. All authors took part in finalizing the manuscript.
