## [Peer Review File · Nature Communications]

Reviewers' comments:

Reviewer #1 (Remarks to the Author):

REVIEW OF "Seasonal modulation of deep slow-slip and earthquakes on the Main Himalayan Thrust" by Panda et al.

This paper considers the effect of seasonal load variations measured by gravity (GRACE) and cGPS measurements on the production of seismicity and aseismic transient in the Himalayan plate boundary.

One of the key results of the paper is shown on Figure 1 and especially 1d where a clear correlation between rainfall or equivalent water height and the large historical earthquakes is observed. The second key result is shown in Figure 2 where a region located below the base of the seismogenic fault zone shows a higher number of slow slip transients.

This is a very interesting paper, well written with clear figures and supplements. I think the authors should have insisted more on the results of Figure 1 (see Major Points). I also think that the fault resonance hypothesis is not demonstrated clearly enough and suggest below an alternative approach.

Anyhow, I think that the observations alone are robust and important enough to be published in Nature Geoscience after some moderate changes that I list below.

Hugo Perfettini

Major Points

- The results of Figure 1d are among the most important in this paper. There is a clear correlation between the rainfall and the historic earthquakes. I think that there are some major questions that could be answered analyzing those data. Is the stacked seismicity rate proportional to the loading rate due to rainfall? Is there a phase lag between the two which could reveal something about the frictional properties of the interface? I guess that other relationships could be explored and I think it is important that the authors discuss more extensively the relationships existing between the physical quantities shown on Figure 1d. That will give the paper a greater impact on the community.
- Why do you assume those values of $a-b$ for the flat and ramp parts? This has to be somehow justified as I think that the difference in response of those two regions to the seasonal variations is essentially due to the difference in $a-b$.
- You give D_c and $a-b$ but to compute T_c , this is not enough and b/a has to be given. Please give the value of b/a that have been assumed.
- If one looks for the highest possible response for resonance at period T , then $k=k_c$ and $T=T_c$ are required. Consequently, I wouldn't be surprised that the lower bound of stiffness considered in the numerical simulations was $k=1.002*k_c$.

An approach that is more appropriate is to look for the $(a-b)$, σ (effective normal stress) and D_c parameters such that $k=k_c$ and $T=T_c=1$ year. If one assumes that $k=c*G/h$, where G is the shear modulus, h the radius of the resonating patch and c a geometric constant of order unity, then $k=k_c$ transforms into $h=h_c$ where $h_c=c*G/k_c$. This implies that the characteristic size of the resonant region should be of the order of h_c . I don't know if the authors could find a parameter set for which $T_c=1$ yr

together with h_c being of the order of the size of the region where they assume that resonance occurs.

If such a set can be found, this could support the idea that a portion of the MHT is resonating with the seasonal variations because its size, frictional properties and normal stress are such that h_c matches the size of this resonant region. If no such parameter set can be found, then probably the resonance mechanism is not responsible for the observed excess of slip transients. This is to me the only viable demonstration of the occurrence of a fault resonance.

Minor Points

- Shear stress higher on the ramp than on the flat part: is this difference only due to the difference in dip angle? Please clarify the possible reasons explaining this difference in shear stress amplitude and especially if it is due to a geometric effect.
- The equation used in the supplement (Fault resonance model) where used by Lowry (2006) but where derived in Perfettini and Schmittbuhl (2001).

Reviewer #2 (Remarks to the Author):

Review,

The manuscript by Panda et al. studied seasonal transient crustal deformation in the Himalayan region. They reported larger GPS seasonal transient deformation than modeled from GRACE and hydrological models, and attributed it to tectonic deep slow slip on the fault ramp of MHT. The authors modeled stress changes by surface load and showed increased shear and Coulomb stress on the ramp of the fault, and also indicated larger velocity perturbation on the MHT ramp than on its flat using fault resonance theory. This is a very interesting discovery, and if some of the results can be further confirmed I may recommend it for publication. I provide my comments below.

Major concern:

The disagreement between GPS measured and GRACE (and hydrological) modeled seasonal deformation in the Himalayan region had been reported and studied before. Chanard et al (2014) attributed it to Earth structure and geocenter motion. Fu et al. (2013) argued that it might be caused by low spatial resolution of GRACE and some local effect, such as groundwater, rivers and so on. This manuscript indicates it is actually tectonic deep slow slip. I believe more evidences are (still) needed to confirm it is tectonics.

Figure S1, there are only 4 GPS stations showing higher and abnormal seasonal transient deformation. Why not all the stations close to the locking zone show higher transient deformation?

I assume the seasonal deformation in this study is an average using data from many years. Is it possible that some stations may only show this kind of tectonic transient slow slip in some specific years? Time-dependent features?

Is it possible to plot the residues (GPS observation - hydrological model)? If this is tectonics, the residues should look like slip-induced surface deformation, both horizontal and vertical, which may (or may not) be different from loading deformation?

I guess the slip-induced horizontal deformation may be different from loading horizontal deformation in terms of horizontal moving direction. I am not sure how much difference it is. It may be too small to separate.

Fig 3, is the stress and velocity perturbation large enough to cause deep slow slip? More discussions?

Minor changes:

L20: can be significantly influenced.

L24: by seasonal hydrological loading deformation.

L40: temporal evolution in Southeast Asia.

L63: ... strain accumulation, the physics of ...

L69: deformation in Nepal Himalaya. A point ...

L94-95: "it is not surprising that ...", this is not a formal expression in a scientific paper.

L114: seasonal hydrological deformation.

L143: sensitive to fault geometry?

Do you show GRACE modeled loading deformation?

Reviewers' comments:

Reviewer #1 (Remarks to the Author):

REVIEW OF "Seasonal modulation of deep slow-slip and earthquakes on the Main Himalayan Thrust" by Panda et al.

This paper considers the effect of seasonal load variations measured by gravity (GRACE) and cGPS measurements on the production of seismicity and aseismic transient in the Himalayan plate boundary.

One of the key results of the paper is shown on Figure 1 and especially 1d where a clear correlation between rainfall or equivalent water height and the large historical earthquakes is observed.

The second key result is shown in Figure 2 where a region located below the base of the seismogenic fault zone shows a higher number of slow slip transients.

This is a very interesting paper, well written with clear figures and supplements. I think the authors should have insisted more on the results of Figure 1 (see Major Points). I also think that the fault resonance hypothesis is not demonstrated clearly enough and suggest below an alternative approach.

Anyhow, I think that the observations alone are robust and important enough to be published in Nature Geoscience after some moderate changes that I list below.

Hugo Perfettini

Reply: Thank you for your valuable suggestions and positive comments. We have addressed each of your queries and have provided line by line replies below.

Major Points

- The results of Figure 1d are among the most important in this paper. There is a clear correlation between the rainfall and the historic earthquakes. I think that there are some major questions that could be answered analyzing those data. Is the stacked seismicity rate proportional to the loading rate due to rainfall? Is there a phase lag between the two which could reveal something about the frictional properties of the interface? I guess that other relationships could be explored and I think it is important that the authors discuss more extensively the relationships existing between the physical quantities shown on Figure 1d. That will give the paper a greater impact on the community.

Reply: Thanks for your suggestions for additional analysis. According to your suggestion, now we have analyzed the cross-correlation and possible phase lag among the physical parameters (i.e. rainfall, equivalent water height, micro-seismicity, current seismicity, and historic earthquake and their corresponding rates) shown in Fig.1d. In that context, we are enclosing some additional figures (Fig S1-S4) below.

Now, we have discussed these new observations, including cross-correlation results in the revised MS, by providing emphasis on various physical parameters. We introduced five new supplementary figures to document this analysis (Supplementary Fig.1- Fig.5).

Figure S1:(Left panel) Stacked panels of monthly rainfall in the region surrounding Nepal , historic earthquake frequency (1255-2015 of $M>6$),micro-seismicity from Nepal (1995-2002 of $M>3$), current earthquake frequency from USGS catalog (1963-2015 of $M>5$) and USGS catalog seismic moment (M_o , Nm) in a region between latitude 25° and 30.5°N and Longitude 75° - 88°E . (Right panel) Cross-correlation between rainfall and various seismic parameters. The yellow bar in the left panel marks the period of the seasonal loading deformation.

Inference: The cross-correlation between the rainfall and historic earthquake is high (~ 0.8) with no phase lag. Rainfall/Micro-seismicity cross-correlation is high (~ 0.7) with a phase difference of 3 months. Rainfall/USGS EQs cross-correlation is significantly less compared to others. However, cross-correlation between the rainfall and seismic moment of earthquakes from USGS catalog is significant with value ~ 0.8 , and insignificant phase difference of ~ 1 month.

Figure S2: (Left panel) Stacked panels of equivalent water height (derived from GRACE averaged over the region latitude: 20°-27.5°N, longitude: 75°-88°E), historic earthquake frequency (1255-2015 of M>6), microseismicity from Nepal (1995-2002 of M>3), current earthquake frequency from USGS (1963-2015 of M>5) and USGS catalog seismic moment (Mo, Nm). (Right panel) Cross-correlation between EWH and monthly seismicity number (and seismic moment from USGS current seismicity). The yellow bar marks the timing of the seasonal deformation.

Inferences: EWH/historic earthquake shows cross-correlation value of ~0.7, with phase difference of 2 months. EWH/micro-seismicity cross-correlation shows a phase lag of 3-4 months with correlation value of ~0.6. Cross-correlation between EWH and earthquakes from USGS catalog is significantly less (~0.4). Finally, EWH/USGS seismic moment cross-correlation shows a time lag of ~2 months and with good correlation value (~0.7). Note that, the phase difference in the cross correlations with EWH is usually ~2 months as compared to that while cross-correlating with rainfall. This is because, there is an inherent phase difference of ~1.5-2 months between the rainfall and EWH.

Figure S3: (Left panel) Stacked panels of equivalent water height gradient/rate, historic earthquake frequency (1255-2015 of $M>6$), micro-seismicity from Nepal (1995-2002 of $M>3$), current earthquake frequency from USGS (1963-2015 of $M>5$) and USGS catalog seismic moment (M_o , Nm). (Right panel) Cross-correlation between EWH rate and monthly seismicity number (and seismic moment from USGS current seismicity). The yellow strip marks the timing of the seasonal deformation.

Inferences: These results are similar to Fig.R1. EWH rate/historic earthquake cross correlation has good correlation value (~ 0.7) and a phase difference of 2 months. EWH rate/micro-seismicity cross correlation has slightly low value (0.6) with a phase difference of $\sim 3-4$ months. Cross correlation between EWH rate and USGS EQs is significantly less (~ 0.35). EWH rate/USGS seismic moment cross correlation shows good correlation value of ~ 0.7 , and has a time lag of 1 month.

Figure S4: (Left panel) Stacked panels of equivalent water height gradient/rate and monthly gradient of event rates (DN/Dt) of micro-seismicity from Nepal (1995-2002 of $M>3$), current earthquake frequency from USGS (1963-2015 of $M>5$) and USGS seismic moment rate (DMo/Dt). (Right panel) Cross-correlation between EWH rate and monthly rate of seismicity number (and seismic moment rate from USGS current seismicity). The yellow bar marks the timing of the seasonal deformation.

Observations: EWH rate/micro-seismicity rate cross correlations has a time lag of $\sim 3-4$ months, with correlation value of ~ 0.5 . Cross correlation between EWH rate and USGS seismicity rate is insignificant (~ 0.3). EWH rate/ USGS seismic moment rate cross correlation shows moderate correlation value of 0.4 and a time lag of 2 months.

Fig. S5: cross-correlation between GRACE derived EWH and rainfall. Note the obvious time lag of about 1.5 month (rainfall lagging EWH), with strong correlation value of ~ 0.75 . EWH is related to the cumulative rainfall and hence its peak appears once the rainfall seasonal is over.

From the above analysis (Fig. S1-S5), it is inferred that the GRACE derived EWH and regional precipitation has strong correlation (~ 0.7) with the occurrence of historic earthquakes and with the seismic moments of USGS earthquake catalog. The correlation is moderate for small earthquakes and weak or no correlation with micro-earthquakes.

Line No: 63-69

• Why do you assume those values of a-b for the flat and ramp parts? This has to be somehow justified as I think that the difference in response of those two regions to the seasonal variations is essentially due to the difference in a-b.

Reply: Thanks for pointing this.

The value of frictional parameter $(b-a) = 10^{-8}$ to 10^{-6} has been proposed for the bottom of the seismogenic zone at hydrothermal conditions (Lowry, A.R., 2006; Blanpied et al., 1995, Foster et al., 2013), which can be considered representative of the environment of the Himalayan ramp. On the other hand, $(b-a) = 0.005$ is considered for intermediate depths of the seismogenic zone (Lowry A.R., 2006; Blanpied et al., 1995); corresponding to the Himalayan flat. Representative frictional parameter $(b-a)$ values used in this work are generally compatible with the laboratory estimates of frictional parameters (Perfettini et al., 2001; Lowry, A.R., 2006; Blanpied et al., 1995; Foster et al., 2013). Now we have highlighted this in the revised MS.

The mid-crustal seismicity belt in the Nepal Himalaya also coincides with the zone of high electrical conductivity of trapped fluids along the MHT, which possibly comes from metamorphic dehydration of the underthrusting Indian basement rocks (Lemonnier et al., 1999; Avouac, 2003). From Fig.S6, it appears that with the lowering of effective normal stress (or increasing pore fluid pressure), there is an increase in the amplitude of velocity perturbation process (i.e., fault resonance process). Therefore, the effect of the fault resonance process can also be influenced by the presence of fluids, in response to variations in pore fluid pressure.

Now we have added this new observation in the revised MS.

Line No: 178-181

Fig. S6: Increase in amplitude of velocity perturbation (i.e., fault resonance process) with increase in pore-fluid pressure (or lowering effective normal stress).

• You give D_c and a-b but to compute T_c , this is not enough and b/a has to be given. Please give the value of

b/a that have been assumed.

Reply: For intermediate depths of the seismogenic zone (i.e., the Himalayan flat), the value of 'b-a' is assumed to be 0.005, which is consistent with the laboratory estimates of frictional parameters (Perfettini et al., 2001; Blanpied et al., 1995). Further, the value of 'a' is considered to be 0.005 (e.g., Perfettini et al., 2001). Therefore, the value of b/a for Himalayan flat region is assumed to be ~2 (or $a/b = 0.005/0.01 = 0.5$). However, for the regimes where slow-slip occurs, the ratio a/b is expected to exceed 0.9 (Foster et al., 2013). We acknowledge that there is little or no constraint on these values for the MHT.

We have included this point in the revised MS.

Line No: 168

• If one looks for the highest possible response for resonance at period T, then $k=k_c$ and $T=T_c$ are required. Consequently, I wouldn't be surprised that the lower bound of stiffness considered in the numerical simulations was $k=1.002*k_c$.

An approach that is more appropriate is to look for the (a-b), sigma (effective normal stress) and D_c parameters such that $k=k_c$ and $T=T_c=1$ year. If one assumes that $k=c*G/h$, where G is the shear modulus, h the radius of the resonating patch and c a geometric constant of order unity, then $k=k_c$ transforms into $h=h_c$ where $h_c=c*G/k_c$. This implies that the characteristic size of the resonant region should be of the order of h_c . I don't know if the authors could find a parameter set for which $T_c=1$ yr together with h_c being of the order of the size of the region where they assume that resonance occurs.

If such a set can be found, this could support the idea that a portion of the MHT is resonating with the seasonal variations because its size, frictional properties and normal stress are such that h_c matches the size of this resonant region. If no such parameter set can be found, then probably the resonance mechanism is not responsible for the observed excess of slip transients. This is to me the only viable demonstration of the occurrence of a fault resonance.

Reply: Thanks for your comment and suggestion.

In the manuscript we have suggested that the mid-crustal ramp is sensitive to the fault resonance process under periodic hydrological loading. Further, the rate-and-state frictional model also provides scaling arguments for faults to slip by the resonance process caused by periodic stress oscillations (Foster et al., 2013). In that case, the half-length of the minimum available patch size required to nucleate such slip event, h^* , is given by (Rice, 1993; Rubin, 2008): $h^* = \frac{D_c}{b-a}$, moreover, h^* is similar to parameter h_c proposed by Perfettini et al., (2001).

Here, G' is the effective shear modulus ($G' = G/(1-\nu)$), G is the shear modulus and ν is the Poisson's ratio. Considering the values of $G = 30$ GPa, $\nu = 0.25$, $b-a = 10^{-6}$ and critical slip distance (D_c) = 10^{-6} m and effective normal stress (σ_e) as 300 MPa for the Himalayan ramp segment, the minimum available patch size required to nucleate a slip event (h^*) is found to be ~0.133 km. However, upon decreasing the effective normal stress to 30 MPa (or increase in pore fluid pressure), the dimension of h^* is increased to ~1.33 km.

Further, for slow slip governed solely by rate-and-state friction to remain stable, the fault length must be smaller than $2L$ (Rubin and Ampuero, 2005), where $L = \frac{D_c}{b-a}$. Now, with the above mentioned parameters, L is found to be ~424 km, assuming effective normal stress (σ_e) as 300 MPa. Moreover, L will increase as we decrease the effective normal stress (or increase in pore fluid pressure).

Finally, it has been suggested that the condition required for a portion of the fault to slip through fault resonance process should follow $2h^* < W < 2L_\infty$ (Foster et al., 2013), where W is the length of the slow-slip patches defined by; $W = \frac{2h^*}{\beta}$, where β is an empirical constant of the order of 1, h^* is slip amount, and v is the slip speed during the slow slip event.

However, in order to investigate fault resonance process in the Himalaya, it is difficult to constrain the exact values of u_{SSE} and V_{SSE} . Therefore, considering that limitation, if we assume that the whole length of Himalayan mid-crustal ramp is resonating, then in that case W would be ~ 25 km. This indicates that the above-mentioned condition (i.e. $2h^* < W < 2L_\infty$) for nucleation of slow-slip events and associated slip perturbations is generally consistent for the Himalayan ramp region.

We have discussed this observation in the revised MS.

Line No: 183-205

Minor Points

- Shear stress higher on the ramp than on the flat part: is this difference only due to the difference in dip angle? Please clarify the possible reasons explaining this difference in shear stress amplitude and especially if it is due to a geometric effect.

Reply: Thanks for pointing this. Yes, the higher shear stress value on the mid-crustal ramp than the flat is mainly due to the difference in dip of the ramp and flat segments (ramp= 20° and flat 7°).

- The equation used in the supplement (Fault resonance model) where used by Lowry (2006) but where derived in Perfettini and Schmittbuhl (2001).

Reply: Thanks for the correct reference. Now we have cited the actual reference in the revised MS for Fault resonance model.

Line No: 545

Reviewer #2 (Remarks to the Author):

Review,

The manuscript by Panda et al. studied seasonal transient crustal deformation in the Himalayan region. They reported larger GPS seasonal transient deformation than modeled from GRACE and hydrological models, and attributed it to tectonic deep slow slip on the fault ramp of MHT. The authors modeled stress changes by surface load and showed increased shear and Coulomb stress on the ramp of the fault, and also indicated larger velocity perturbation on the MHT ramp than on its flat using fault resonance theory. This is a very interesting discovery, and if some of the results can be further confirmed I may recommend it for publication. I provide my comments below.

Reply: Thank you for your valuable suggestions. We have addressed each of your comments and provide line by line replies below.

Major concern:

The disagreement between GPS measured and GRACE (and hydrological) modeled seasonal deformation in the Himalayan region had been reported and studied before. Chanard et al (2014) attributed it to Earth structure and geocenter motion. Fu et al. (2013) argued that it might be caused by low spatial resolution of

GRACE and some local effect, such as groundwater, rivers and so on. This manuscript indicates it is actually tectonic deep slow slip. I believe more evidences are (still) needed to confirm it is tectonics.

Reply: We appreciate the concern of the reviewer. We do agree that Chanard et al. (2014) and Fu et al. (2013) provided some explanations for the transients. In both cases, the amplitude of transients should decrease as we move towards the north, i.e., away from the source. The poor resolution and role of Earth structure variability certainly can lead to some disagreement between the observed and computed transient motions, but these are mostly in phase, not in magnitude, including in the horizontal components. These articles do not explain the increase in transients seen in the ramp region of Himalaya, which are reported here (Fig.1). Such a systematic variation in the ramp region implies that it has to be governed by some localised process occurring in that region. We have described these previous results in more detail and contrast them with our finding of substantially larger amplitudes of the horizontal seasonal motions for a subset of stations near the down-dip ramp region of the MHT mentioned in lines 110-117 in the MS.

Figure S1, there are only 4 GPS stations showing higher and abnormal seasonal transient deformation. Why not all the stations close to the locking zone show higher transient deformation?

Reply: We agree that there are only 6 sites (SYBC, HRMN, CHLM, DRCL, BYNA, TPLJ) which show high transients. One of the key findings of the work is that the presence of the mid-crustal ramp causes increase in shear stress which results in resonance leading to high transients in the GPS time series. Thus, the regions where there is no ramp or the frictional parameters are different from those needed to produce the resonance phenomena, such high abnormal transients would not be expected. In fact if our hypothesis is correct and if we have sufficiently dense coverage of cGPS in the future, the presence of these high transients in the GPS time series may act as a proxy for the presence of a ramp with the requisite frictional properties in that segment (see Figure S1).

In addition to that, the mid-crustal seismicity belt in the Nepal Himalaya also coincides with the zone of high electrical conductivity of trapped fluids along the MHT, which possibly come from metamorphic dehydration of the underthrusting Indian basement rocks (Lemonnier et al., 1999; Avouac, 2003). From Fig.S1, it appears that with the lowering of effective normal stress (or increasing pore fluid pressure), there is an increase in amplitude of the velocity perturbation (i.e., fault resonance process). Therefore, effect of the fault resonance process can also be influenced by the presence of fluid, in terms of variation in pore fluid pressure.

Fig.S1: Increase in amplitude of velocity perturbation (i.e., fault resonance process) with increase in pore-fluid pressure (or lowering effective normal stress).

Now we have added this new observation as supplementary figure (Supplementary Figure 9) in the revised MS.

Line No: 178-181

I assume the seasonal deformation in this study is an average using data from many years. Is it possible that some stations may only show this kind of tectonic transient slow slip in some specific years? Time-dependent features?

Reply: cGPS sites which exhibit high transients show such behavior every year since their installation (some sites are more than 10 years old). So we do not think that the abnormal horizontal transients at a site are a time dependent phenomenon. In fact if our hypothesis is correct then it implies that it should not be a time dependent feature. Below we show example time series (DRCL, CHLM).

Fig. R1: Detrended GPS time series of DRCL and CHLM, which show abnormally high transients values. [Editorial Note: Figure R1 reproduced from Johnson, K. M., Shelly, D. R. & Bradley, A. M. Simulations of tremor-related creep reveal a weak crustal root of the San Andreas Fault. Geophys. Res. Lett. 40, 1300–1305 (2013), with permissions from John Wiley and Sons. All rights reserved.]

Is it possible to plot the residues (GPS observation - hydrological model)? If this is tectonics, the residues should look like slip-induced surface deformation, both horizontal and vertical, which may (or may not) be different from loading deformation?

Reply: Thanks for suggesting this. According to your suggestion we have plotted the residues (i.e. GPS observation – hydrological model) of the two cGPS sites (CHLM and DRCL) which exhibit high seasonal transients. The residues for the horizontal components appear similar to slip-induced surface deformation and are consistent with the GPS observation. However, in case of vertical component for the two sites, there is a net uplift in the residuals during the loading period, possibly hydrological load induced deep-slow slip at mid-crustal ramp. Based on this context, we are providing a figure below.

Fig. R2: Monthly stacked time series of GPS observations, hydrological load model and the residuals (GPS observation – hydrological model) of CHLM and DRCL. Note, there is an elevated residuals value in the vertical component during the loading period.

So in conclusion, we find that the relatively large transients (particularly in the north) at some of the sites in the ramp region are much larger than the effect of the hydrological load and could be driven by the hydrological load induced slow slip in the ramp region.

Further, one of the difficulties to determine the residuals at each GPS sites is the lack of resolution and accuracy in the global hydrological models. The magnitude at most of the sites, which do not show high transients match but there are several sites where there is a phase difference. The phase difference is the most prominent in the horizontal components as the direction of horizontal displacement is dependent on the relative position of the site with respect to the hydrological load distribution. If the load distribution does not have sufficient resolution and accuracy, the resulting horizontal displacement would be quite different than actually observed on the cGPS time series. Gautam et al. (2017) also highlighted this at a site BDRI in the Garhwal region of NW Himalaya.

Fig. R3: The three panels above show the N, E and Up components of GPS displacement time series (Gautam et al., 2017). DELI is in the Indo-Gangetic plains, WIHG is in Outer Himalaya, GHUT and PITH are in Lesser Himalaya and BDRI is in the Higher Himalaya. Red curve is the estimated displacement due to global hydrological model. Note the inconsistency in phase at BDRI in the N and E component.

I guess the slip-induced horizontal deformation may be different from loading horizontal deformation in terms of horizontal moving direction. I am not sure how much difference it is. It may be too small to separate.

Reply: The above response also applies here. At sites which do not show any abnormal transients, the difference is not much. Moreover, we are providing an enlarged monthly stacked time series of two sites showing high transient values (DRCL and CHLM), to show the difference between the loading horizontal deformations and horizontal moving direction.

Fig. R4: Enlarged monthly stacked time series of CHLM and DRCL, along with hydrological and atmospheric loads.

Fig 3, is the stress and velocity perturbation large enough to cause deep slow slip? More discussions?

Reply: Thanks for suggesting this, now we discuss this question and Fig. 3 in more detail in the revised MS.

In the manuscript we have suggested that the mid-crustal ramp is sensitive to fault resonance process, due to periodic hydrological loading. Further, rate-and-state frictional model also provide scaling arguments for faults to slip by resonance process caused by periodic stress oscillations (Foster et al., 2013). In that case, half-length of the minimum available patch size required to nucleate such slip event, h^* , is given by (Rice, 1993;

Rubin, 2008): $h^* = \frac{G' D_c}{(b-a)\sigma_e}$ Moreover h^* is the like parameter h_c proposed by Perfettini et al., (2001).

Where G' is the effective shear modulus ($\frac{G}{1-\nu}$), G is the shear modulus and ν is the Poisson's ratio.

Considering the values of $G=30$ GPa, $\nu=0.25$, $b-a=10^{-6}$ and critical slip distance (D_c) as 10^{-6} m and effective normal stress (σ_e) as 300 MPa for the Himalayan ramp segment, the minimum available patch size required to nucleate a slip event (h^*) obtained as ~ 0.133 km. However, upon decreasing the effective normal stress to 30 MPa (or increase in pore fluid pressure), the dimension of h^* is increased to ~ 1.33 km.

Further, for a slow-slip governed solely by rate-and-state friction to remain stable, the fault length must be smaller than $2L_\infty$ (Rubin and Ampuero, 2005), where $L_\infty = \frac{G' D_c b}{\pi \sigma_e (b-a)^2}$. Now, with the above mentioned parameters, L_∞ is found to be ~ 424 km, assuming effective normal stress (σ_e) as 300 MPa. Moreover L_∞ will increased as we decrease the effective normal stress (or increase in pore fluid pressure).

Finally, it has suggested that the condition required for a portion of the fault to slip through fault resonance process should follow $2h^* < W < 2L_\infty$ (Foster et al., 2013), where W is the length of the slow-slip patches defined by: $= \frac{u_{SSE} G'}{\beta \sigma_e (b-a) \ln(V_{SSE}/V_{creep})}$, where β is an empirical constant of order 1, u_{SSE} is slip amount, and V_{SSE} is the slip speed during the slow slip event.

However, in the present study in order to investigate fault resonance process in Himalaya, it is difficult to constraints the exact values of u_{SSE} and V_{SSE} . Therefore, considering that limitation, if we assume that the whole length of Himalayan mid-crustal ramp is resonating, then in that case W would be ~25 km. This indicates that, the above mentioned condition (i.e. $2h^* < W < 2L_\infty$) for nucleation of slow-slip events and associated slip perturbations is quiet consistent for Himalayan ramp region.

We have included this discussion in the revised manuscript.

Line No: 183-205

Minor changes:

L20: can be significantly influenced.

Done

Line No: 21

L24: by seasonal hydrological loading deformation.

Done

Line No: 24

L40: temporal evolution in Southeast Asia.

Done

Line No: 41

L63: ... strain accumulation, the physics of ...

Done

Line No: 70

L69: deformation in Nepal Himalaya. A point ...

Done

Line No: 75

L94-95: "it is not surprising that ...", this is not a formal expression in a scientific paper.

Done

Line No: 100

L114: seasonal hydrological deformation.

Done

Line No: 120

L143: sensitive to fault geometry?

Do you show GRACE modeled loading deformation?

Reply: Seasonal hydrological load induced normal, shear and coulomb stress change are computed on the Himalayan ramp (dip=20°) and flat (dip=7°) segments. The stress changes are sensitive to the geometry of the fault (i.e. dip). That's why the mid-crustal ramp exhibits higher values of coulomb stress perturbations.

Yes, the modelled loading deformation and associated stress change on the Himalayan ramp and flat is derived from hydrological load, in terms of equivalent water height (EWH), derived from the GRACE observations.

Done

Review of “Seasonal modulation of deep slow-slip and earthquakes on the Main Himalayan Thrust” by Panda et al.

The authors have answered to most of my comments. However, I still have a concern concerning the modeling part.

To my original question:

“Why do you assume those values of a-b for the flat and ramp parts? This has to be somehow justified as I think that the difference in response of those two regions to the seasonal variations is essentially due to the difference in a-b

”

The authors have answered

« Reply: Thanks for pointing this.

The value of frictional parameter $(b-a) = 10^{-8}$ to 10^{-6} has been proposed for the bottom of the seismogenic zone at hydrothermal conditions (Lowry, A.R., 2006; Blanpied et al., 1995, Foster et al., 2013), which can be considered representative of the environment of the Himalayan ramp. On the other hand, $(b-a) = 0.005$ is considered for intermediate depths of the seismogenic zone (Lowry A.R., 2006; Blanpied et al., 1995); corresponding to the Himalayan flat. Representative frictional parameter $(b-a)$ values used in this work are generally compatible with the laboratory estimates of frictional parameters (Perfettini et al., 2001; Lowry, A.R., 2006; Blanpied et al., 1995; Foster et al., 2013). Now we have highlighted this in the revised MS.

The mid-crustal seismicity belt in the Nepal Himalaya also coincides with the zone of high electrical conductivity of trapped fluids along the MHT, which possibly comes from metamorphic dehydration of the underthrusting Indian basement rocks (Lemonnier et al., 1999; Avouac, 2003). From Fig.S6, it appears that with the lowering of effective normal stress (or increasing pore fluid pressure), there is an increase in the amplitude of velocity perturbation process (i.e., fault resonance process). Therefore, the effect of the fault resonance process can also be influenced by the presence of fluids, in response to variations in pore fluid pressure.

Now we have added this new observation in the revised MS. »

Lowry (2006) indeed wrote “Near the base of the Guerrero seismogenic zone, where events most probably initiate, such a calculation yields $a \sim 0.015$ and $(b-a) \sim 10^{-9}$. These values approximate laboratory estimates of frictional parameters at the base of the seismogenic zone [Blanpied et al., 1995]. »

It is incorrect to state that Blanpied et al. (1995) found such small values of b-a. Page 13,061, Appendix A, Blanpied et al. (1995) proposed values of b-a of the order of - 0.0025. Experimentally, the smallest values of a or b that can be found are of the order of 10^{-4} (Chris Marone, personal communication) so that values of b-a lower than 10^{-4} are not supported experimentally. In Foster et al. (2013), the values proposed are $b-a=1.5 \times 10^{-4}$ and are sound from an experimental point of view, although in the limit of what is resolvable experimentally.

If the authors can find parameters such that $h=h_c$ and $T=T_c$ with values of $b-a$ larger than 10^{-4} , then the resonance phenomenon can be considered as viable. If not, then this proposed mechanism will appear as purely speculative.

If the authors can answer positively to this request, then the paper will be suitable for publication.

Hugo Perfettini

Reviewers' comments:

Review of “Seasonal modulation of deep slow-slip and earthquakes on the Main Himalayan Thrust” by Panda et al.

The authors have answered to most of my comments. However, I still have a concern concerning the modeling part.

To my original question:

“Why do you assume those values of $a-b$ for the flat and ramp parts? This has to be somehow justified as I think that the difference in response of those two regions to the seasonal variations is essentially due to the difference in $a-b$ ”

The authors have answered

« Reply: Thanks for pointing this.

The value of frictional parameter $(b-a) = 10^{-8}$ to 10^{-6} has been proposed for the bottom of the seismogenic zone at hydrothermal conditions (Lowry, A.R., 2006; Blanpied et al., 1995, Foster et al., 2013), which can be considered representative of the environment of the Himalayan ramp. On the other hand, $(b-a) = 0.005$ is considered for intermediate

depths of the seismogenic zone (Lowry A.R., 2006; Blanpied et al., 1995); corresponding to the Himalayan flat. Representative frictional parameter (b-a) values used in this work are generally compatible with the laboratory estimates of frictional parameters (Perfettini et al., 2001; Lowry, A.R., 2006; Blanpied et al., 1995; Foster et al., 2013). Now we have highlighted this in the revised MS.

The mid-crustal seismicity belt in the Nepal Himalaya also coincides with the zone of high electrical conductivity of trapped fluids along the MHT, which possibly comes from metamorphic dehydration of the underthrusting Indian basement rocks (Lemonnier et al., 1999; Avouac, 2003). From Fig.S6, it appears that with the lowering of effective normal stress (or increasing pore fluid pressure), there is an increase in the amplitude of the velocity perturbation process (i.e., fault resonance process). Therefore, the effect of the fault resonance process can also be influenced by the presence of fluids, in response to variations in pore fluid pressure. Now we have added this new observation in the revised MS. »

Lowry (2006) indeed wrote “Near the base of the Guerrero seismogenic zone, where events most probably initiate, such a calculation yields $a \sim 0.015$ and $(b-a) \sim 10^{-9}$. These values approximate laboratory estimates of frictional parameters at the base of the seismogenic zone [Blanpied et al., 1995]. »

It is incorrect to state that Blanpied et al. (1995) found such small values of b-a. Page 13,061, Appendix A, Blanpied et al. (1995) proposed values of b-a of the order of -0.0025. Experimentally, the smallest values of a or b that can be found are of the order of 10^{-4} (Chris Marone, personal communication) so that values of b-a lower than 10^{-4} are not supported experimentally. In Foster et al. (2013), the values proposed are $b-a = 1.5 \times 10^{-4}$ and are sound from an experimental point of view, although in the limit of what is resolvable experimentally.

Reply:

Thanks for your comment on our specific choice of frictional parameters (b-a). We do agree that the laboratory measurements of average estimate of (b-a) value for velocity weakening conditions are larger in magnitude and are of the order of 10^{-4} , which is experimentally supported (Foster et al., 2013, GRL). Thanks for correcting this.

However, the slip sensitivity (or fault resonance process) gets larger as (b-a) approaches smaller values. In case of velocity strengthening (i.e., stable sliding) the (b-a) value should be less than zero, while for velocity weakening (unstable sliding) the condition is opposite, i.e., (b-a) should be positive. However, near the exact frictional transition from velocity strengthening to velocity weakening, which is the region of occurrence of slow slip events (in this case, the Himalayan mid-crustal ramp), there should be a transition of (b-a) value from negative to positive (He et al., 2007; Lowry, 2006, Scholz, 1998; Johnson et al., 2013). Therefore, near the exact frictional transition zone, the values of (b-a) can become vanishingly small near where it passes through zero (Prof. Tony Lowry, Department of Geology, Utah State University, personal communication, July 2018). Therefore, assuming near-neutral rate dependence near

the transition we have selected a very low (b-a) value for the Himalayan mid-crustal ramp of the order of $\sim 10^{-8}$.

We acknowledge that such extremely low values of (b-a) have not actually been determined in laboratory experiments, but they reflect the range and variation of values presented in He et al. (2007). Now we have corrected this argument in the revised manuscript. But we argue that the exact implementation of this laboratory based estimate of frictional parameter on the natural condition may not be straight forward due to wide range and uncertainty in estimation. Further, He et al., 2013, JGR (doi:10.1002/jgrb.50280) has suggested that how even small changes in conditions and composition (here addition of a few % quartz) can flip small (b-a) values in the hydrothermal transition regime. Here is a summary Figure R1 from Johnson and Shelly (2013, GRL) showing the range of (a-b) parameters from Gabbro (He et al., 2007) and Granite (Blanpied et al.1995) at varying depths and temperatures at hydrothermal conditions, similar to Liu and Rice (2009, JGR). There is clearly a wide range and uncertainty in using any quantitative estimate of (a-b).

Figure 5. Range of laboratory values of *a-b* for gabbro [He et al., 2007] and wet granite [Blanpied et al., 1995] and range of values inferred in this study. Approximate temperature-to-depth conversion is based on Sass et al. [1997]. Black bars extend across entire range of reported laboratory values in the two studies.

Figure.R1

Finally, we suggest that under natural conditions, near the transition of velocity weakening to velocity strengthening, the values of (b-a) should indeed be very small. This argument is supported by Lowry (2006, Nature), where he suggested that the slip sensitivity gets larger as (b-a) decreases but remains positive, which will always be the case near the frictional transition from velocity strengthening to velocity weakening; which is why SSEs are most commonly observed near the basal frictional transition.

In the view of above, we have revised our manuscript and we hope that the revised manuscript will meet your favorable recommendation for publication.

Line No: 166-173

REVIEWERS' COMMENTS:

Reviewer #1 (Remarks to the Author):

The authors have answered satisfactorily to all my comments. Therefore, I recommend the paper to be accepted in its present form.

Hugo Perfettini